# The Basic Principles of Pathophysiology of Venous Thrombosis

**DOI:** 10.3390/ijms252111447

**Published:** 2024-10-24

**Authors:** Sam Schulman, Alexander Makatsariya, Jamilya Khizroeva, Victoria Bitsadze, Daredzhan Kapanadze

**Affiliations:** 1Department of Medicine, Thrombosis and Atherosclerosis Research Institute, McMaster University, 1280 Main St W, Hamilton, ON L8S 4L8, Canada; 2Department of Obstetrics, Gynecology and Perinatal Medicine, The I.M. Sechenov First Moscow State Medical University (Sechenov University), Trubetskaya Str 8-2, 119435 Moscow, Russia; gemostasis@mail.ru (A.M.); jamatotu@gmail.com (J.K.); vikabits@mail.ru (V.B.); 3Center of Pathology of Pregnancy and Hemostasis «Medlabi», Tbilisi 340112, Georgia; daka.kapanadze@yandex.ru

**Keywords:** venous thromboembolism, hypoxia, stasis, cytokines, neutrophil extracellular traps, complements

## Abstract

The past few decades have brought tremendous insight into the molecular and pathophysiological mechanisms responsible for thrombus generation. For a clinician, it is usually sufficient to explain the incident of deep vein thrombosis (DVT) with provoking factors such as trauma with vascular injury, immobilization, hormonal factors, or inherited or acquired coagulation defects. About half of DVTs are, however, lacking such triggers and are called unprovoked. Venous stasis and hypoxia at the valve sinus level may start a chain of reactions. The concept of immunothrombosis has added a new dimension to the old etiological triad of venous stasis, vessel wall injury, and changes in blood components. This is particularly important in COVID-19, where hyperinflammation, cytokines, and neutrophil extracellular traps are associated with the formation of microthrombi in the lungs. To better understand the mechanisms behind DVT and reach beyond the above-mentioned simplifications, animal models and clinical epidemiological studies have brought insight into the complex interplay between leukocytes, platelets, endothelium, cytokines, complements, and coagulation factors and inhibitors. These pathways and the interplay will be reviewed here, as well as the roles of cancer, anticancer drugs, and congenital thrombophilic defects on the molecular level in hypercoagulability and venous thromboembolism.

## 1. Introduction

The pathophysiology of venous thrombosis has historically been explained with Virchow’s triad, which consists of hypercoagulability, venous stasis, and injury to the endothelium of the blood vessels. Although the physician-scientist Rudolf Virchow studied venous thrombosis with its consequences and coined the terms venous thrombosis and pulmonary embolism, he did not explicitly state that the components of the triad were the cause of venous thrombosis, as subsequently explained [1]. Only about a hundred years later, Virchow’s triad started being quoted [2]. Hypercoagulability can occur in patients with malignancy, inflammation, increased levels or increased activity of specific coagulation factors (factor V, factor VIII, fibrinogen), thrombocythemia, polycythemia, dehydration, disseminated intravascular coagulation, heparin-induced thrombocytopenia and its variant, and catastrophic antiphospholipid syndrome. Venous stasis is seen during immobilization, external obstruction as with a tourniquet, during total knee replacement, internal obstruction by a tumor or an enlarged organ (uterus during pregnancy) compressing a vein, or endovascular obstruction (indwelling catheter). Changes in the vessel wall include mechanical endothelial injury (by catheters or surgical interventions) and invasive cancer, or secondary to stasis with leukocyte adhesion.

Studies in animal models have improved our understanding of the pathophysiology of thrombosis. The events on a molecular level during venous stasis have been investigated. Furthermore, the effects of hypoxemia have evoked substantial interest. Whole-genome sequencing and other molecular–genetic analyses have led to new important discoveries regarding thrombotic mechanisms. Although Rudolf Virchow ridiculed a French colleague for proposing that inflammation was causative in thrombosis [1], we now know better. Thromboinflammation has become a major player in the field.

## 2. The Basic Principles of the Pathophysiology of Venous Thrombosis

### 2.1. Venous Stasis

Situations when venous stasis is implicated in the generation of thrombosis include deep vein thrombosis (DVT) in the lower extremity during immobilization, left atrial appendage thrombus formation in atrial fibrillation, left ventricle thrombosis due to akinetic myocardium and infarction or due to actual obstruction of the blood flow such as external compression from an expansive process (typically a tumor), or from restricted venous lumen in catheter-related arm vein thrombosis or possibly also in recurrent DVT where there is residual thrombus or fibrotic scar from a prior event. On a molecular level, there are a few possible mechanisms: hypoxemia, activated coagulation factors, and inflammatory reaction. In a dog model, as well as in patients, it has been shown that the blood within the pocket of the venous valves, also called valve sinus, became rapidly hypoxic during stasis, and thrombus formation on the valve cusp could be demonstrated after 2 h of non-pulsatile flow [3].

#### 2.1.1. Hypoxemia 

Yan et al. at Nigel Mackman’s laboratory used a murine lung thrombosis model and concluded that since an antibody against tissue factor (TF) prevented fibrin deposition during hypoxia, the expression of TF is an important component of the process [4]. Likewise, depletion of monocytes reduced fibrin accumulation in the hypoxic lung. The transcription factor early growth response-1 (Egr-1) seems to be an important modulator of the TF expression because in homozygous-Egr-1-null mice, there was no increase in TF antigen and no fibrin deposition during hypoxia [5]. The same research group subsequently demonstrated that protein kinase C (PKC) beta-isoform is upstream to Egr-1 since PKC-beta-null mice showed only minimal Egr-1 increase during hypoxia [6]. Following this, Lo et al., in Taiwan, found, using a bovine aortic endothelium model, that PKC alpha is also a mediator, and they further demonstrated that the signaling pathway goes from hypoxia triggering a temporary translocation of PKC alpha from cytosol to a membrane fraction, where it associates with Ras or Raf-1 or ERK1/ERK2 and eventually induces Egr-1 [7].

Another pathway is via the hypoxia-inducible factor-1α (HIF-1α), which normally is hydroxylated and degraded, but under hypoxic conditions, it becomes stabilized and able to mediate transcriptional activation of various genes, including those involved in angiogenesis, apoptosis, and metastases, and this is mediated by TF [8].

On the other side of the hemostatic mechanisms is fibrinolysis; plasminogen activator inhibitor-1 (PAI-1) is a major factor for suppressing fibrinolysis and thereby increase the risk of thrombosis. Liao et al., in Ann Arbor, Michigan, demonstrated in a murine macrophage model that the increase in PAI-1 in response to hypoxia is an effect of both increased transcription and improved mRNA stability [9]. This PAI-1 upregulation is mediated by HIF-1α (possibly the dominant pathway), Egr-1 and CCAAT/enhancer binding protein alpha.

There are also reports of increased levels of factors VIII, VIIa, XII, fibrinogen, endothelial protein C receptor, Annexin IV, thrombin-activatable fibrinolysis inhibitor, antiplasmin and platelet activation, as reviewed by Bikov et al. [10].

The main hypoxia-induced pathways leading to increased thrombus formation are summarized in Figure 1.

High-altitude-associated pulmonary embolism has been reported in at least two different contexts. In a retrospective study of 31,581 patients undergoing 1- to 2-level posterior lumbar fusion at a hospital attitude of >5000 feet above sea level and as many matched patients who had the same surgery at <100 feet altitude, the rate of pulmonary embolism was 0.49% versus 0.35%, resulting in an odds ratio (OR) of 1.38 (95% confidence interval [CI] 1.08–1.76) with no significant difference in DVT [11].

In a systematic literature review of sleep apnea and thrombosis, 15 studies were identified and in 14 of those, sleep apnea was an independent risk factor for DVT or for pulmonary embolism [12]. The risk for VTE was, as shown in two prospective studies, 2- to 3-fold higher compared to that for persons without sleep apnea [12]. A recent health care database study in France reported that the time spent with oxygen saturation below 90% was an independent predictor for unprovoked VTE with a hazard ratio of 1.06 (95% CI, 1.01–1.02), or in other words, the more severe nocturnal hypoxia, the higher the risk for VTE [13]. Although ascent to high altitudes has been associated in several case reports and case series with the development of pulmonary embolism, a systematic review of two studies was inconclusive regarding connection between hypoxia and thromboembolic events, and the authors speculated that exercise load, training status, mental stress and fluid status could be confounding factors in this context [14]. They found in the studies that explored hemostatic variables that there was increased thrombin generation but also decreased platelet activation, and at altitudes above 5400 m, there was activation of coagulation and at the same time increased fibrinolysis. Whether hypoxemia in the arteries transfers to hypoxemia in the veins is questionable, and therefore the role of arterial hypoxemia for generation of DVT can be doubted.

#### 2.1.2. Reduced Clearance in Valve Pockets 

Brooks et al. reported in 2009 that the endothelium from the valvular sinus in the greater saphenous vein has higher levels of the endothelial protein C receptor (EPCR) and thrombomodulin and lower expression of von Willebrand factor (VWF) than in endothelium in the rest of this vein [15]. This implies that, under normal conditions, there is a dominant anticoagulant environment in the valve sinus. However, in a subsequent study from the same group, there was not a significant difference in the expression of EPCR or thrombomodulin between endothelium in the valvular sinus versus in parts of the vein; thus, only VWF levels differed [16]. The increase in VWF levels in the non-valvular endothelium was between 3-fold and 6.7-fold in different individuals.

It can then be hypothesized that, during venous stasis, there is a change in the balance towards hypercoagulability. In a mouse model with stenosis causing an 80–90% reduction in the venous lumen, Brill et al. and von Brühl et al. reported that there was activation of the endothelium with the release of VWF and P-selectin [17,18]. The latter binds to P-selectin glycoprotein ligand-1 on leukocyte, and VWF binds to the GP1bα receptor on platelets that then become activated. The leukocytes, in turn, become activated and express more tissue factor. When the venous circulation is impaired, it is quite possible that these coagulation factors, activated platelets and leukocytes fail to clear and instead promote thrombin generation locally.

#### 2.1.3. A Genetic Component

In a study with whole-exome sequencing, followed by gene-based collapsing analysis, 400 patients with VTE were compared with a large, general population cohort, and rare variants associated with VTE were identified [19]. As could be expected, three of the variants encoded the well-known natural anticoagulants protein C, protein S and antithrombin. The 4th gene was STAB2, which stood out due to having more than four times as many rare coding variants among patients with VTE compared to the controls. The corresponding protein is Stabilin-2, which is a scavenger receptor on the endothelial surface. The investigators found that all the patients with VTE and Stabilin-2 variants had impaired intracellular transport, for example, of VWF. Indeed, individuals with STAB2 variants in an independent cohort had elevated VWF levels, which can contribute to thrombus formation.

### 2.2. Venous Flow Pattern

The blood flow in veins is characterized by a low shear rate, which is typically about 100/s [20]. The flow rate near the venous wall influences how molecules are transported to and from a surface that may have become reactive. Activation of coagulation is more likely to happen the lower the flow rate and the larger the affected endothelial area are. Fibrinogen and possibly other plasma proteins enhance the formation of bridges between cells under very low shear stress, such as in postcapillary venules. This results in rouleaux formation of red cells and in certain clinical conditions in three-dimensional erythrocyte aggregates, which play a major role in vascular resistance in these venules. At shear rates below 100/s, interactions occur between erythrocytes, platelets, and neutrophils, mediated by intercellular adhesion molecule-4 (ICAM-4) [21]. Thereby, leukocytes become more prone to colliding with the venous wall, resulting in an added increase in flow resistance. The local increase in hematocrit increases the margination of platelets. Thus, a very low shear rate can, via increasing resistance, result in stagnation of the flow.

On the other hand, narrowing of the venous lumen, as occurs with the use of compression stockings, results in faster blood flow, reduced flow separation and lower risk of thrombosis.

### 2.3. Interplay Between Blood Components and Endothelium

The initiation of thrombus formation happens when factor VIIa, together with TF, expressed on monocytes, activates factor X to Xa, which, in turn, with factor Va as cofactor on a phospholipid bilayer (monocyte or endothelium), activates factor II (prothrombin) to IIa (thrombin). At this early stage, only small amounts of thrombin are formed, insufficient for cleavage of fibrinogen to fibrin. Instead, the thrombin molecules that have been formed (1) split the VWF-factor VIII complex to generate factor VIIIa, (2) activate factor XI to XIa, (3) activate factor V to Va, and (4) activate platelets. These alterations shift the coagulation cascade into high gear. Factor XIa activates factor IX to IXa, which, together with factor VIIIa, speeds up the conversion of factor X to Xa, which, in turn, together with more of factor Va, generates large amounts of thrombin to cleave off fibrinopeptide a and b from fibrinogen. The hereby formed fibrin monomers polymerize to fibrin threads that trap platelets and red cells on the endothelial surface. More TF is then delivered with extracellular vesicles that may have been released from activated platelets or injured endothelium, allowing the clot to continue to grow. Components that promote clot formation are (1) collagen, exposed to the blood stream at the site of injured endothelium, (2) endothelin—the most potent vasoconstrictor in our cardiovascular system—released from activated endothelium, (3) VWF released from Weibel–Palade bodies in the endothelium, and (4) PAI-1, also released from the endothelium, and that is incorporated in the clot to prevent premature fibrinolysis.

### 2.4. Inflammation

This is the more recent addition to Virchow’s triad and an area of intensive research. A large number of components have been implicated in the mechanism of thrombus formation, adding several layers of complexity to the old coagulation cascade. This review will focus on the roles of interleukins, neutrophil extracellular traps, the complement system, and a possible genetic risk factor.

#### 2.4.1. Interleukins and Their Possible Role in Air-Pollution-Evoked Venous Thromboembolism

The above-described nuclear transcription factor Egr-1 regulates the expression of proteins such as interleukin (IL) 1β and CXCL2 that are involved in inflammatory processes. IL1β was demonstrated to be elevated in inflamed colon tissue, which in this study was experimental colitis, and when IL1β was injected to control animals, it dose-dependently enhanced thrombus formation [22]. Furthermore, IL1β levels were higher in patients with COVID-19 that developed thrombotic events than in those without such complication [23]. In fact, a number of different ILs have been described to play a role in thrombus generation. The CD4+ Th17 cells produce IL9 and IL17A, which increase platelet aggregation rate, the expression of P-selectin and the development of thrombosis or acute coronary syndrome [24,25]. IL6 was recently reported to be elevated in cerebral venous sinus thrombosis and it was deemed useful for differentiating the condition from anatomical variants [26]. IL6 was also shown to be independently associated with high FVIII, which is a well-described risk factor for thrombosis [27]. IL18 was elevated in patients with ankle fracture who developed DVT compared to those who did not [28].

The mechanism of increased VTE risk from air pollution also seems to be mediated by IL-6. In a mouse model with intratracheal instillation of 10 μg of particulate matter with diameter smaller than 10 μm (PM_10_), there was activation of coagulation, as measured by shortened bleeding time, prothrombin time and activated partial thromboplastin time [29]. This was explained by increased levels of fibrinogen and coagulation factor II, VIII and X, whereas there was no platelet activation, as assessed with flow cytometry. The IL-6 level in bronchoalveolar fluid was also increased, but when macrophages were depleted, the response to particulate matter and the thrombotic tendency were reduced. Furthermore, when IL-6 knockout mice were exposed to the particulate matter, there was no shortening of the bleeding time or evidence of activation of coagulation.

Alternative pathophysiological mechanisms have also been implicated in air-pollution-mediated inflammation and thrombosis. Mast cells and the release of histamine seem to play a role in air-pollution-associated hypercoagulability, as shown in a hamster model [30]. When 50 μg of diesel exhaust particles was instilled into the trachea, there was airway inflammation and release of histamine in broncho-alveolar lavage and in plasma. A small venous thrombosis, induced before the exposure, increased in size. Pretreatment with dexamethasone prevented these alterations. A similar study was performed in 20 human volunteers who were exposed to 350 mg/m^3^ of dilute diesel exhaust [31]. Soluble P-selectin and soluble CD40 ligand increased significantly after 6 h compared to after inhalation of filtered air, which served as control, but there was no significant increase in IL-6 or C-reactive protein. In this study, there was evidence of platelet activation, based on increased platelet–neutrophil and platelet–monocyte aggregates in flow cytometry. Thrombus formation was assessed ex vivo in a Badimon chamber and it increased significantly by 27% at 2 h after the inhalation of diesel exhaust compared to inhalation of filtered air.

These experimental studies are supported by an epidemiological study in Santiago, Chile [32]. Hospitalizations for DVT or pulmonary embolism in 2001–2005 were compared against local data on air pollution from O_3_, NO_2_, SO_2_, CO, PM_10_ and PM_2.5_. There were significant associations between hospitalization for VTE—stronger for PE than for DVT—and increased levels of these air pollutants. A systematic literature review identified 11 studies that overall suggested an association between particulate matter and VTE, but due to great variability between the studies in methodology, measurement of effects and duration of follow-up, the authors were unable to perform a meta-analysis to provide quantitative data [33].

#### 2.4.2. Neutrophil Extracellular Traps

Activated platelets have been described to release inorganic polyphosphate (polyP), which, in turn, activates factor XII and thereby the contact activation pathway, leading to fibrin formation but also the kallikrein–kinin system with release of bradykinin, resulting in inflammatory reactions [34]. Neutrophils that become activated by damaged endothelium or by activated platelets will then release tissue factor as well as neutrophil extracellular traps (NETs) [35]. NETs consist of a network of DNA strings and associated histones and a number of neutrophil granule proteins, e.g., elastase, lactoferrin, myeloperoxidase and α1-antitrypsin. Charles Esmon’s group reported that histones have cytotoxic effects on endothelium in vitro, and when administered in vivo, the histones caused vacuolated endothelium as well as micro- and macrovascular thrombosis in mice [36]. Co-administration of histones with activated protein C prevented death and when the protein C activation was blocked, death occurred with sublethal doses of histones. 

Nucleosomes consist of DNA segments coiled around a core of histone and are derived from NETs. In 149 patients that were diagnosed with DVTs, nucleosomes and elastase-α1-antitrypsin complexes were elevated compared to 183 controls, consisting of patients with suspected DVT but negative ultrasound examination [37]. The median nucleosome level in patients with DVT was 17 U/mL (interquartile range [IQR] 9–35), and in controls, it was 9 U/mL (IQR 5–17), *p* < 0.001, and the level of elastase-α1-antitrypsin complexes was 53 ng/mL (IQR 43–71) in those with DVT and 45 ng/mL (IQR 33–55) in controls, *p* < 0.001. 

Myeloperoxidase–DNA complexes are generated during the formation of NETs and can serve as a biomarker for NETs. A recent study demonstrated elevated plasma levels of myeloperoxidase–DNA complexes in patients with arterial as well as with venous thrombosis, although an increased level was not predictive of future cardiovascular events or death [38]. 

NETs can activate the intrinsic pathway directly via the activation of factor XII or via the release of polyP from trapped and thereby activated platelets. A possible sequence of events, starting from impaired venous flow and involving NET formation, is shown in Figure 2. Furthermore, leukocyte elastase, released from activated neutrophils, is known to cleave tissue factor pathway inhibitor (TFPI) by proteolysis of the polypeptide linking Kunitz-1 with Kunitz-2 domain, and thereby promote the extrinsic pathway of coagulation [39]. Activated platelets and endothelial cells release protein disulphide isomerase, which can alter the tissue factor conformation to allow for decryption and activation and further enhance the extrinsic pathway of coagulation [40].

These prothrombotic responses are collectively termed thromboinflammation (sometimes interchangeably with immunothrombosis), which implies a pathophysiologic condition with injured vascular endothelium, impaired anti-inflammatory and antithrombotic functions. Thromboinflammation is even more pronounced in severe sepsis, trauma, transplant rejection, and ischemia-reperfusion injury. Recently recognized is the thromboinflammatory process in patients with severe coronavirus disease 2019, with pronounced inflammatory response to acute lung injury and development of microthrombosis in pulmonary blood vessels [41].

In COVID-19, the viral proteins with pathogen-associated microparticles (PAMPs) cause systemic hyperinflammation that may lead to excessive immune response with release of interleukins, interferons and tumor necrosis factor α—“cytokine storm” [42]. The high levels of IL8 promote the recruitment of neutrophils and release of NETs. Damaged alveolar cells and lymphocytes release histones, which have been found in circulation in high concentrations in patients with COVID-19. Histones activate platelets and coagulation, either free in the circulation or as components of the NETs, where microthrombi have been shown to form. Serum biomarkers of NETs, including citrullated histones, nucleosomes, DNA, neutrophil elastase and myeloperoxidase are seen in increased levels in patients with COVID-19 and thrombosis, and are associated with poor outcome, as reviewed by Xu et al. [43]. There is now high-quality evidence that therapeutic-dose heparin in patients hospitalized for COVID-19 not only prevents thromboembolism but also reduces mortality [44]. 

#### 2.4.3. Complement System 

In the context of damaged endothelium, neutrophils are recruited and activated, resulting in the release of various proteolytic enzymes and NETs and a complex cross-talk between the complement and contact coagulation pathways as recently reviewed [45]. This cross-talk between the complement system and coagulation represents another mechanism for VTE that has received attention recently. In a prospective Danish study with over 80,000 persons, complement C3 levels were higher already at baseline among those that developed VTE during follow-up (n = 1176) than in those that did not; the hazard ratio for the third tertile of C3 was 1.58 (95% CI, 1.33–1.87) [46]. Subsequently, Hansen’s group in Tromsø, Norway, showed that increased levels of the terminal complement complex C5b-9 predicted unprovoked VTE with an OR in the highest quartile of C5b-9 of 1.74 (95% CI, 1.10–2.78) compared to age- and sex-matched controls [47]. The same group also reported an association between C5 and future VTE risk and this became stronger for unprovoked versus provoked VTE [48]. They subsequently found that an overactive complement system due to insufficient regulation in patients with C1 inhibitor deficiency, who suffer from hereditary angioedema, was also associated with increased risk of VTE. Additionally, in humans as well as in a mouse model, there was an increase in prothrombin fragment 1 + 2, thrombin-antithrombin complex and thrombin generation, and in the mice, there was also more frequent VTE but not arterial thrombosis [49].

#### 2.4.4. Genetic Susceptibility

In a genome-wide association scan of samples from 5862 patients with a history of venous thrombosis and 7112 controls, the *HIVEP1* locus (chromosome 6p24.1) was identified as a susceptibility locus for the disease [50]. The corresponding protein binds to DNA sequences in the promoter and enhancer regions of inflammatory target genes, resulting in transcriptional regulation. The presence of the *HIVEP1* rs169613C allele demonstrated an association with increased VTE risk (OR 1.20; 95% CI, 1.13–1.27). 

### 2.5. Cancer and Thrombosis

#### 2.5.1. Risk Factors of Cancer-Associated Thrombosis

Patients with active cancer have a 20% risk of being diagnosed with VTE and an additional proportion of patients are found to have occult thrombosis or pulmonary embolism on autopsy [51,52]. Thrombosis is the second most common cause of death in patients with cancer [53], and if a patient with cancer develops VTE, the prognosis and survival is worse than in patients without cancer [54]. Cancer also increases the risk of recurrent VTE and at the same time the risk of bleeding complications [55].

There is a multitude of risk factors associated with VTE in patients with cancer. At a high level, they can be divided into patient-related, treatment-related and cancer-related risk factors [56]. The patient-related factors are typically the same as in patients without cancer, such as older age and immobility. As for the treatment-related factors, surgery with immobilization and tissue damage and central venous catheters with partial obstruction of the vein are not much different either from patients without cancer. The tumor- and treatment-related thrombogenic mechanisms are summarized in Figure 3.

#### 2.5.2. Pharmaceutical Treatments for Cancer and Risk of Thrombosis

**Chemotherapeutic agents** have been used alone or in combinations for cancer therapy for about 80 years, with nitrogen mustard as the first agent. This was followed by the development of antifolates, such as methotrexate, which does not appear to have any toxic effect on endothelium [57]. In the 1950s, the development of antimetabolites started with 6-mercaptopurine and 5-fluorouracil (5-FU). The latter has been associated with cardiotoxicity and coronary thrombosis [58] or left ventricle thrombosis [59]. Cwikiel et al. showed that when 5-FU was incubated with endothelial cells, DNA synthesis decreased significantly [57]. The same group demonstrated that 5-FU could disrupt the endothelial cell barrier, thereby exposing the subendothelial matrix, with ensuing platelet aggregation [57,60]. It has, however, not been convincingly shown that treatment with 5-FU is associated with an increased risk of VTE. 

The anthracyclines are thrombogenic due to several effects, which include the activation of coagulation, as evidenced by an increase in D-dimers and thrombin-antithrombin complexes, and this increase could be prevented with a prophylactic dose of LMWH [61]. A research group at McMaster University, Canada, led by Patricia Liaw, showed that in endothelial cells that were exposed to doxorubicin, there was down-regulation of the mRNA of endothelial protein C-receptor (EPCR) and EPCR shedding, resulting in decreased EPCR levels on the endothelial surface and reduced activation of protein C [62]. They then demonstrated that anthracyclines increased the exposure of phosphatidylserine and expression of TF on endothelial cells as well as on monocytes [63], which was also shown by Nigel Mackman’s group [64]. Subsequently, they found that anthracyclines, as well as 5-FU, given to healthy mice increased the release of cell-free DNA, which correlated with an increase in thrombin–antithrombin complexes and in thrombin generation [65]. Others have reported that patients treated with anthracyclines have decreased vasomotor reactivity, implying that the endothelial function was impaired compared to controls [66]. All these prothrombotic alterations have clinical importance, since in a study of patients with multiple myeloma receiving two different chemotherapy regimens, one of which included doxorubicin, the latter was associated with a 6.5-fold increase in VTE [67]. L-asparaginase reduces the level of antithrombin, and the development of VTE had already been reported in 1979 [68]. A meta-analysis of 17 studies including 1752 patients treated with L-asparaginase reported a VTE incidence of 5.2% (95% CI: 4.2–6.4) [69]. Jacqueline Conard and others studied the mechanism for this and demonstrated that there is a decrease in antithrombin, protein C, protein S, FIX, FX, prothrombin and fibrinogen in plasma, with the natural anticoagulant reduction dominating initially, generating this prothrombotic state [70,71,72].

The anticancer properties of cisplatin were accidentally discovered and the agent was introduced clinically in the 1970s. The mechanism of action of cisplatin and its analogs, carboplatin, oxaliplatin and nedaplatin, is through crosslinking of DNA, resulting in apoptosis of the tumor cells. Evidence of a procoagulant activity of cisplatin was demonstrated in 1990 based on TF-like activity from exposed monocytoid cells [73]. Later, Lechner et al. found that cisplatin caused the release from endothelium of highly procoagulant microparticles, but the ensuing thrombin generation was driven by phospholipids rather than TF [74]. Other reported procoagulant effects of cisplatin are the upregulation of FXa and increased formation of thrombin on platelets [75]. A meta-analysis of 38 trials with 8216 patients showed that chemotherapy that included cisplatin increased the risk of VTE more than two-fold to 1.92% compared to 0.79% in controls, and the risk seems to be dose-dependent [76]. 

**Hormonal agents** for cancer therapy were developed based on early observations of regression of breast cancer and prostate cancer after removal of the ovaries and testicles, respectively. Thus, in 1941, Huggins and Hodges initiated treatment of patients with prostate cancer with orchiectomy or estrogen [77]. The goal was to achieve an anti-androgen effect, with luteinizing hormone-releasing hormone (LHRH) synthetic agonists, gonadotropin-releasing hormone antagonists (GnRHs) and, in order to block the adrenal androgen production, the use of steroidal or non-steroidal antiandrogens, typically in combination [78]. Hormonal therapies are also very effective for receptor-positive breast cancer, starting with tamoxifen, first produced in 1962, and with the mechanism of action through its anti-estrogenic effect in breasts, although it has estrogenic effects in the uterus and liver. It is thus a selective estrogen receptor modulator (SERM). The effects of tamoxifen on hemostasis are, similar to those of estrogen, prothrombotic, with a reduction in the natural anticoagulant levels (antithrombin, protein C, total protein S) and increase in coagulation factor activity (factors VIII, IX, VWF) [79], and increased activated protein C resistance [80]. Tamoxifen also contributes to platelet activation via the activation of phospholipase Cγ and phosphoinositide-3-kinase, which results in the release of intracellular free Ca^2+^ from the endoplasmatic reticulum [81]. Aromatase inhibitors prevent the enzyme from converting androgen into estrogen in adipose tissue and are also used for the management of breast cancer. Anastrozole and letrozole belong to the family of selective aromatase inhibitors. A meta-analysis of 25 studies found that compared to the VTE prevalence of 0.5% in the general population, aromatase inhibitors had a VTE prevalence of 2.95%, although that was lower than for tamoxifen, OR 0.61 (95% CI, 0.37–1.00), driven by lower risk for DVT (OR 0.68) but not for pulmonary embolism (OR 1.01) [82]. Although there is no evidence of aromatase inhibitor-associated changes in the levels of coagulation factors and inhibitors [83], an in vitro study showed that anastrozole decreased P-selectin expression and induced platelet aggregation and fibrin network formation [84]. 

**Immunomodulators** include thalidomide, lenalidomide and pomalidomide. Although none of these agents seem to have any effect on coagulation on their own, when given together with dexamethasone for treatment of multiple myeloma, CD62P increased on platelets and the closure time in the Platelet Function Analyzer-100 shortened, and increased levels of F VIII and soluble thrombomodulin were observed [85]. In a meta-analysis of 3322 patients with multiple myeloma, treatment with thalidomide, dexamethasone and the combination increased the risk of VTE 2.6-, 2.8- and 8-fold, respectively, whereas concomitant prophylactic dose anticoagulation eliminated that increase in risk [86]. The VTE risk is similar with second-generation immunomodulators. 

**Growth factors** that are targeted with monoclonal antibodies for cancer therapy include epidermal growth factor and its receptor (EGFR) and vascular endothelial growth factor (VEGF, **anti-angiogenic agents**). A meta-analysis of 17 studies on the anti-EGFR agents, cetuximab and panitumumab, showed a 50% increase in the risk of VTE compared to those receiving other regimens [87], but the mechanism for hypercoagulability has not been elucidated. The risk of VTE is higher when these anti-EGFR drugs are used in combination with some other chemotherapeutic agents such as cisplatin- or irinotectan-derivatives. Similarly, the recombinant monoclonal antibody necitumumab, which blocks the EGFR ligand binding site, increased VTE risk by about 60% when added to other chemotherapy in two different trials [88,89]. Although the VEGF-directed antibody bevacizumab was shown to increase the expression of PAI-1 and the thrombus development in femoral vein injury or inferior vena cava obstruction in a mouse model [90], the use of bevacizumab in 10 randomized clinical trials or of aflibercept in a meta-analysis did not reveal an increased risk of VTE [91,92].

**Kinase inhibitors** include receptor tyrosine kinase inhibitors (RTKI) and cyclin-dependent kinase inhibitors (CDK). Whereas the first generation TKI, imatinib, was not associated with increased risk of VTE, the second generation agents, the breakpoint cluster region protein-tyrosine kinase protein ABL1 derivatives, such as ponatinib, increased the expression of VWF and platelet adhesion with microvascular angiopathy [93]. In a meta-analysis of randomized controlled trials, second-generation TKIs (ponatinib, nilotinib, dasatinib) were mainly associated with arterial thromboembolism, but there was also a 3-fold increase in venous occlusive events compared to imatinib [94]. Of the CDK inhibitors, it seems that mainly abemaciclib is associated with an increased risk of VTE (absolute risk increase approximately 4%), but the pathophysiology behind this effect has not been revealed [95]. Trametinib, belonging to the MEK inhibitor class, blocks downstream signaling from MEK1 and MEK2 and thereby inhibits the growth of melanoma cells with BRAF mutation. It improves the effectiveness of BRAF inhibitors, such as dabrafenib, but increases the risk of pulmonary embolism at least 4-fold compared to BRAF monotherapy, as seen in a systematic review and meta-analysis [96]. The mechanism for this thrombogenic effect is unclear, although nitric oxide (NO) pathway inhibition has been suggested. The reasoning is that inhibition of BRAF and MEK causes the upregulation of cluster of differentiation 47, thereby inhibiting the signaling of NO-cyclic guanosine monophosphate and diminished bioavailability of NO with ensuing vasoconstriction and hypercoagulability [96].

Selpercatinib, an inhibitor of mutated forms of RET tyrosine kinases and used for lung and thyroid cancers, was linked with increased incidence of pulmonary embolism, DVT, and pericardial effusion in a review of the FDA adverse events’ reporting system [97].

#### 2.5.3. Cancer Related Factors

Adenocarcinoma that secretes mucin seems to be more thrombogenic than, for example, squamous cell carcinoma [98]. These abnormally glycosylated mucins carry binding sites for selectins and interact with L-selectins (on leukocytes) and P-selectins (on platelets), resulting in platelet aggregation and thrombin generation [99]. Important selectin ligands on tumor cells and secreted mucin are sialyl-Lewis (SLe) SLex and SLea [100]. 

Cancer procoagulant (CP) is a cysteine proteinase from malignant tissues which can directly activate F X to Xa [101], and it also promotes cancer metastasis. CP is expressed in a wide range of tumor cells.

Tissue factor is overexpressed in many types of cancer, particularly in pancreatic cancer, and plays a major role in the progression of cancer. The mechanism for TF in cancer-associated thrombosis is probably via the shedding of TF-rich extracellular vesicles from the tumor cells [102]. Neutrophils release more NETs into the circulation in patients with cancer, and these can activate platelets and can synergistically, together with TF-bearing extracellular vesicles assembling on a negatively charged phospholipid surface, create a thrombin burst that not only causes thrombosis but may also promote angiogenesis and tumor metastasis, as reviewed by [103].

Finally, the tumor can by compression of a vein or by intravascular invasion cause stasis and thrombus formation.

### 2.6. Hereditary Thrombophilia

In general, hereditary thrombophilia causes increased risk of VTE by loss of control of thrombin generation or by compromised inhibition of thrombin.

Antithrombin deficiency was the first described hereditary disorder associated with increased risk of thrombosis [104]. The prevalence in the general population is 0.02%. Antithrombin inhibits serine proteases, including kallikrein, factors IXa, Xa, XIa, XIIa and thrombin by forming a stoichiometric 1:1 complex. Proteoglycans on the endothelial surface or heparin speed up this reaction 1000-fold. Due to the effect on so many coagulation factors, antithrombin is the most clinically important of the natural coagulation inhibitors, and homozygous type I deficiency (absence of activity and antigen) is not compatible with life.

Protein C cleaves, in its active form (APC), factors Va and VIIIa, which thereby become inactivated. Protein C deficiency was described in 1981 [105] and is prevalent in 0.2% of the general population. It leads to increased activity of the cofactors Va and VIIIa.

Protein S is a cofactor for APC and enhances localized complex with factor Va and VIIIa on negatively charged endothelial or platelet phospholipid surfaces. However, protein S also has functions independent of APC and can, under certain circumstances, act as an anticoagulant. However, hereditary deficiency of protein S, first described in 1984 [106,107], increases the risk of VTE and is found in 0.03–0.13% of the general population.

Resistance to APC was identified in patients with VTE in 1993 [108]. In 1984, the G1691A mutation, leading to Arg506Gln (R506Q) replacement in factor V and explaining the APC resistance, was reported [109] and was found to be the most common hereditary thrombophilia in the Caucasian population with a prevalence of 3–7%. Other more rare mutations causing APC resistance have subsequently been described in various populations.

Hyperprothrombinemia, associated with a G20210A mutation on the non-translated prothrombin gene region, was described in 1996 [110] and has a prevalence in the Caucasian population of 0.7–4%, with a higher number in the Mediterranean region.

High factor VIII levels are invariably acquired, but in 2021, two Italian families with a severe tendency to VTE were described to have a partial 23.4 kb *F8* gene duplication, hereafter named FVIII Padua [111]. The proband had a factor VIII level of >400%.

## 3. Future Research

The presentation of DVT differs between patients. Why is embolization to the lungs only found in about half of the patients? Why do some develop hormone-associated thrombosis in the leg and others in the cerebral venous sinus? What factors decide when the thrombus stops growing? Why do some develop post-thrombotic syndrome or pulmonary hypertension? These are important clinical questions, but they can only be answered by further studies to better understand the pathophysiologic mechanisms.

## 4. Conclusions

The principles of pathophysiology of VTE have become much more complex than the classic Virchow’s triad. Venous stasis with hypoxia or accumulation of activated coagulation factors, platelets, leukocytes and activated endothelium may initiate inflammatory processes that play a major role in thrombus generation. The latter may involve the release of interleukins, NETs, and activated complement factors. In thromboinflammation, there is evidence of activation of both the extrinsic and intrinsic pathways, as well as reduced inhibition due to the cleavage of TFPI. During the past decade, several genetic variants have been reported to have an association with the risk for VTE, and some of those findings have increased our understanding of additional mechanisms for thrombosis. Rather than assigning a single mechanism to the generation of thrombosis, there is even in the most straight-forward cases of DVT a complex chain of pathophysiologic events involving cell cooperation and multiple biochemical pathways. Fortunately, prevention or treatment of VTE usually only requires inhibition or blockage at one point, such as inhibition of a specific coagulation factor. In more severe forms of thromboinflammation, such as severe sepsis, antiphospholipid syndrome, preeclampsia or Behҁet’s disease, there might be a need for combination therapies to control unrestrained thrombus generation.

Diagnosis, prevention and treatment of VTE and the mechanisms of antithrombotic agents have not been described in this review. Instead we refer the reader to textbooks such as Hemostasis and Thrombosis—Basic principles and clinical practice. Marder VJ, Aird WC, Bennet JS, Schulman S, White GC II. Hemostasis and Thrombosis—Basic principles and clinical practice. Lippincott Williams & Wilkins, Philadelphia 2012: 1-1566.

## Figures and Tables

**Figure 1 ijms-25-11447-f001:**
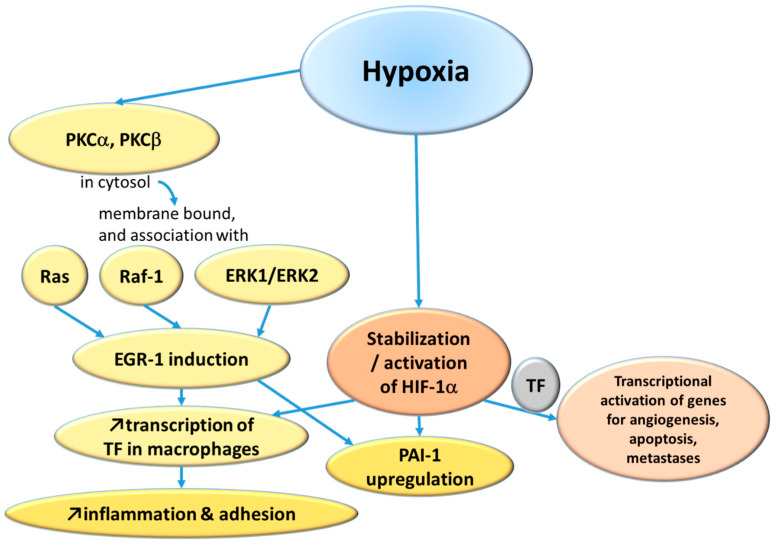
The main hypoxia-induced pathways of thrombus formation. PKC—protein kinase C; ERK—extracellular signal-regulated kinase; EGR—early growth response; HIF—hypoxia-inducible factor; TF—tissue factor; PAI—plasminogen activator inhibitor.

**Figure 2 ijms-25-11447-f002:**
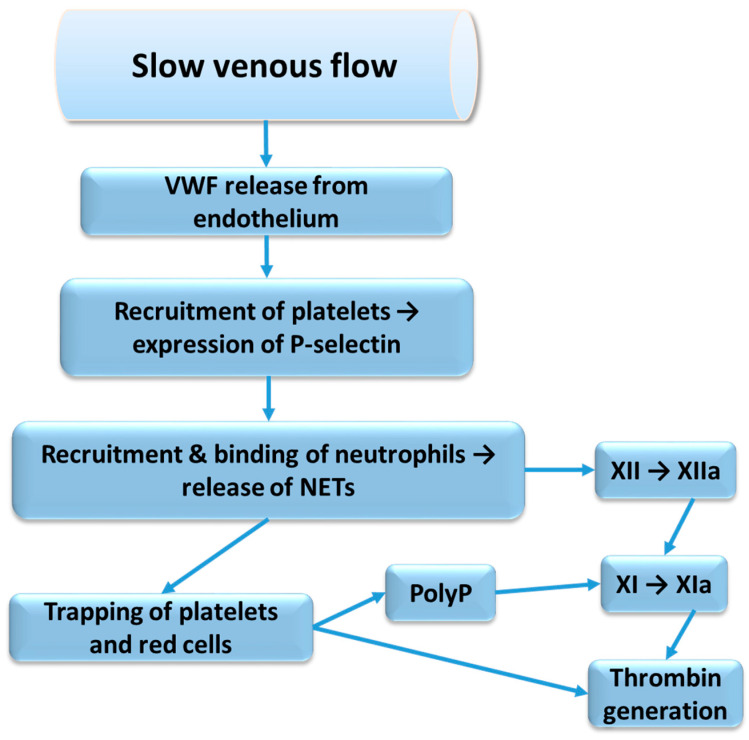
Coagulation pathways’ activation through NETs’ release and platelets’ activation. VWF—von Willebrand factor; NETs—neutrophil extracellular traps; PolyP—polyphosphate.

**Figure 3 ijms-25-11447-f003:**
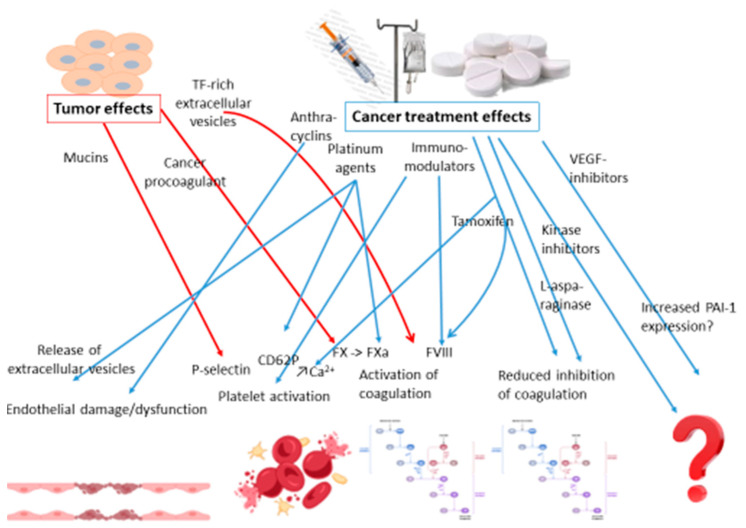
The tumor- and treatment-related thrombogenic mechanisms.

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
