# Peer review of "The Basic Principles of Pathophysiology of Venous Thrombosis"

_ijms, 2024, doi:10.3390/ijms252111447_

Round 1

Reviewer 1 Report

Comments and Suggestions for Authors

1.       While the Virchow’s triad is not the sole explanation of venous thromboembolism, the Authors should describe all three components of the triad. Here, only stasis is adequately addressed. Other two components are indirectly described, it should be changed.

2.       The Authors adequately describe inflammatory processes as an additional mechanism that can be responsible for venous thrombosis. Still, this chapter should be better organized, now it is quite messy

3.       Hypoxemia. Local hypoxemia can be responsible for the initiation of thrombotic cascade, as the Authors describe. Yet, there is a slim evidence of a role for systemic hypoxia (hypoxemia at the arterial side not necessarily transfers to the hypoxemia in the veins), as has been described in the manuscript, however it should be stated more clearly.

4.       The Authors should describe the role of abnormal flow pattern in the veins, such as flow stagnation, separation, etc. Generally, narrowing of the vein is not a risk factor for thrombosis (although it may evoke stasis upstream with subsequent thrombosis there, but not at the site of stenosis). Compression stockings that are recommended as a prophylaxis against thromboembolism result in narrowing of the deep lower extremity veins, consequently the flow in these veins is faster and flow separation is reduced.

5.       The Authors should describe the interplay between blood components and the endothelium in the context of venous thrombosis

6.       Chapter 2.2.1. The title should rather be, for example: “air pollution-evoked venous thromboembolism and possible role for interleukins”

7.       NETs are important in the setting of COVID-associated venous thromboembolism. This issue should be addressed

8.       Thrombosis related to cancer treatment. The Authors should focus at current drugs. Old pharmaceutical agents may be mentioned, yet currently used therapies should be primarily described.

9.       There is almost no information regarding pharmacological and non-pharmacological methods for the treatment and prevention of venous thrombosis, in the context of its pathophysiology.

Reviewer 2 Report

Comments and Suggestions for Authors

Authors reviewed the basic principles of pathophysiology of VTE.

Although this manuscript is potentially interesting, several issues arise.

1.     In genetic component, FV Leiden and AT resistant should be discussed.

2.     Authors should discuss the mechanism of VTE in patients with COVID-19.

3.     Thrombin burst may cause VTE in CAT.

4.     The role of platelet may be important in VTE.

5.     Diagnosis of thrombosis may be helpful for the readers.

6.     Relationship between mechanism of VTE and treatment may be helpful for clinican.

Round 2

Reviewer 1 Report

Comments and Suggestions for Authors

1. References should be given in the format requested by the Journal. In the revised version of manuscript it is not done.

2. Abstract should be rewritten to show the whole text in a "microcosm". Now, for example, the entire part dealing with immunothrombosis is nor present in the abstract.

3. The last sentence could be as follows: Diagnosis, prevention and treatment of VTE and the mechanisms of antithrombotic agents have not been described in this review. Instead we refer the reader to the textbooks, such as
"Hemostasis and Thrombosis – Basic principles and clinical practice (here the bibligraphic data should be given, since textbooks should not be included in the References chapter"

Author Response

Reviewer, 2nd round

Dear Reviewer, I appreciate you reviewing our manuscript. Here are a point-to-point response to the reviewer’s comments:

Comments 1. References should be given in the format requested by the Journal. In the revised version of manuscript it is not done.
Response 1: The publisher has generously offered to take care of reference formatting.

Comments 2. Abstract should be rewritten to show the whole text in a "microcosm". Now, for example, the entire part dealing with immunothrombosis is nor present in the abstract.
Response 2: We have added in the abstract:
“The concept of immunothrombosis has added a new dimension to the old etiological triad of venous stasis, vessel vall injury and changes in blood components. This is particularly important in COVID-19, where hyperinflammation, cytokines and neutrophil extracellular traps are associated with formation of microthrombi in the lungs.”
 And at the end of the Abstract:
“(These pathways and the interplay will be reviewed here,) as well as the roles of cancer, anticancer drugs, and congenital thrombophilic defects on the molecular level in hypercoagulability and venous thromboembolism.”

Comments 3. The last sentence could be as follows: Diagnosis, prevention and treatment of VTE and the mechanisms of antithrombotic agents have not been described in this review. Instead we refer the reader to the textbooks, such as
"Hemostasis and Thrombosis – Basic principles and clinical practice (here the bibligraphic data should be given, since textbooks should not be included in the References chapter"
Response 3: We have changed the text as the reviewer recommended.

Sincerely, 

Sam Schulman